# `BiT`: Robustly Binarized Multi-distilled Transformer

**Zechun Liu**[*]
Reality Labs, Meta Inc.
zechunliu@fb.com

**Barlas Oğuz**[*]
Meta AI
barlaso@fb.com

**Aasish Pappu**
Meta AI
aasish@fb.com

**Lin Xiao**
Meta AI
linx@fb.com

**Scott Yih**
Meta AI
scottyih@fb.com

**Meng Li**
Peking University
meng.li@pku.edu.cn

**Raghuraman Krishnamoorthi**
Reality Labs, Meta Inc.
raghuraman@fb.com

**Yashar Mehdad**
Meta AI
mehdad@fb.com

## Abstract

Modern pre-trained transformers have rapidly advanced the state-of-the-art in machine learning, but have also grown in parameters and computational complexity, making them increasingly difficult to deploy in resource-constrained environments. Binarization of the weights and activations of the network can significantly alleviate these issues, however, is technically challenging from an optimization perspective. In this work, we identify a series of improvements that enables binary transformers at a much higher accuracy than what was possible previously. These include a two-set binarization scheme, a novel elastic binary activation function with learned parameters, and a method to quantize a network to its limit by successively distilling higher precision models into lower precision students. These approaches allow for the first time, fully binarized transformer models that are at a practical level of accuracy, approaching a full-precision BERT baseline on the GLUE language understanding benchmark within as little as 5.9%. Code and models are available at: `https://github.com/facebookresearch/bit`.

## 1 Introduction

The past few years have witnessed tremendous advances in almost all applied fields of AI. It would hardly be a simplification to say that the bulk of these advances was achieved by scaling the transformer architecture (Vaswani et al., 2017) to ever larger sizes with increasing computation budget (Devlin et al., 2019; Liu et al., 2019; Radford et al., 2018, 2019; Raffel et al., 2020; Brown et al., 2020). On the other hand, mobile devices and wearables have proliferated and shrunk in size, with stringent requirements for storage, computation and energy consumption. Consumers demand more portability, while having access to all that current AI technology has to offer. As a result, the gap between what is possible in AI, and what is deployable has never been wider.

While there is a variety of methods to increase inference efficiency in neural networks (e.g. knowledge distillation, pruning), quantization has some attractive properties and has been widely successful in practice (Gholami et al., 2021). For one, storage and latency gains from quantization are deterministically defined for a given quantization level. For instance, reducing the precision of model parameters by a given factor immediately translates to an identical reduction in storage cost. Similarly,

---

[*] Equal contribution

36th Conference on Neural Information Processing Systems (NeurIPS 2022).

reducing the precision of arithmetic operations results in a corresponding reduction in computational cost. Uniform quantization is hardware friendly, making it relatively simple to realize theoretical improvements in practice.

Binarization represents the extreme limit of quantization, promising a $32\times$ reduction in storage over full-precision (32-bit) models. Moreover, binary arithmetic completely eliminates multiplications in favor of bit-wise XNOR operations (Courbariaux et al., 2016; Rastegari et al., 2016), enabling even further improvements when using special purpose hardware. Energy efficiency improvements between 100-1000x have been claimed to be possible with binary neural networks (BNNs), over their full-precision counterparts (Nurvitadhi et al., 2016).

The obvious challenge with binarization is the difficulty of optimization. While all quantization is discontinuous, higher precisions allow approximating the full-precision network to a better extent, where with BNNs, this becomes much harder. Surprisingly, researchers in computer vision have been able to demonstrate BNNs with remarkable accuracy (Liu et al., 2018; Qin et al., 2020; Martinez et al., 2020). Unfortunately, while these works have mostly been developed on convolutional architectures for image tasks, they have not generalized well to transformer models. For instance, recent work (Qin et al., 2021) has shown that a BERT (Devlin et al., 2019) model with binary weights and activations lags its full-precision counterpart by as much as 20 points on average on GLUE dataset. Even for weights-only binarization, the loss landscape was shown to be too irregular, and recent work resorted to complex and specialized methods such as weight-splitting from half-width models to achieve a reasonable accuracy (Bai et al., 2021).

With this background, we tackle the problem of fully binarizing transformer models to a high level of accuracy. With the expansion trend of transformers towards becoming the standard architecture choice for all fields of AI, we believe a solution to this problem could be highly impactful.

Our approach follows the same paradigm as previous work, based on knowledge distillation (Hinton et al., 2015) from higher precision models using the straight-through estimator (STE) of Bengio et al. (2013). In view of the optimization difficulties, we take the following steps to ensure that the student and teacher models are well-matched:

- In Section 3 we describe a robust binarization framework, which allows the binary student network to better match the output distribution of the teacher. This allows us to achieve SoTA results for extreme activation quantization with BERT, producing models with little loss in accuracy down to a quantization level of binary weights and 2-bit activations and improves over previous setups by large margins in the fully binary (1-bit) setting. It also leads to competitive results for weight binarization with 4-bit activations using a single knowledge distillation step.

- To further improve binary models, we propose a multi-distillation approach, described in Section 4. Instead of distilling directly into a fully binary model, we first distill an intermediate model of medium precision and acceptable accuracy. This model then becomes the teacher in the next round of distillation into increasingly quantized models. Such a method ensures that the student model doesn't drift too far from the teacher, while also ensuring as good an initialization as possible. We call the resulting model BiT [2].

In the vanilla setting without data augmentation, our approach reduces the accuracy gap to a full-precision BERT-base model by half on the GLUE (Wang et al., 2019) benchmark compared to the previous SoTA. When using data augmentation, we are able to reduce the absolute accuracy gap to only 5.9 points (from over 15 points previously). In addition to the fully binary setting, we also report SoTA results with binary weights and 2-bit activations, where our models trail the full-precision baseline by only 3.5 points.

## 2 Background

### 2.1 Transformer architecture

The transformer model of Vaswani et al. (2017) is composed of an embedding layer, followed by $N$ transformer blocks and a linear output layer. Each transformer block consists of a multi-head

---

[2]Short for Binarized Transformer.

attention layer followed by a feed-forward network, as shown in Figure 1. The multi-head attention layer is a concatenation of $K$ scaled dot-product attention heads, defined by:

$$\text{Attention}(Q, K, V) = \text{softmax}\left(\frac{QK^T}{\sqrt{d_k}}\right) V$$

where $d_k$ is the dimension of each key, $Q, K, V$ are weight matrices for the query, key and value respectively. As such, the computation in a transformer model is limited to linear matrix multiplications and additions, pointwise non-linearities (most commonly Sigmoid (Han & Moraga, 1995), GeLU (Hendrycks & Gimpel, 2016) or ReLU (Nair & Hinton, 2010) ) and the Softmax operation (Bridle, 1989).

## 2.2 Quantization

A vector $w$ is uniformly quantized to $b$-bit precision, if its entries are restricted to the set $\{0, 1, \ldots, 2^b - 1\}$ for asymmetric case or $\{-2^b, -2^b + 1, \ldots, 2^b - 1\}$ for symmetric case, up to a real-valued scale $\alpha$. This allows vector operations to utilize lower precision arithmetic, making them more efficient by a factor of $\frac{B}{b}$ compared to full-precision calculation using $B$ bits. The scaling operation is still in higher precision, but if the dimensionality of $w \gg \frac{B}{b}$, then the extra computation is negligible.

A neural network with parameters quantized to $b_w$ bits takes up $\frac{B}{b_w}$ times less space. However, to take advantage of lower-precision arithmetic, the input vectors (activations) to each vector/matrix operation also need to be quantized. A network which has weights quantized to $b_w$ bits and activations quantized to $b_a$ bits is denoted as $\text{W}b_w\text{A}b_a$. In this work, we're specifically interested in W1A1 transformers. Binary arithmetic is especially attractive, since multiplications reduce to XNOR operations, and can be implemented orders of magnitude more efficiently using specialized hardware (Nurvitadhi et al., 2016).

## 2.3 Knowledge distillation

Knowledge distillation (KD) (Hinton et al., 2015) is a technique whereby a student network can be trained to mimic the behavior of a teacher network. This is especially useful when the student network is more efficient and easier to deploy than the more complex and cumbersome teacher. The basic way of performing KD is by using the output distribution of the teacher model ($\mathbf{p}$) as soft targets for training the student model. If $\mathbf{q}$ is the student model's output, then we have the loss term:

$$\mathcal{L}_{\text{logits}} = \text{KL}(\mathbf{p}, \mathbf{q}) \tag{1}$$

The advantage of KD over simple supervised training of a more efficient model is that the teacher model provides a richer training signal including model confidence for each output class.

For computer vision tasks, distilling the final logits solely works well for binary neural networks (Liu et al., 2020). If the student and teacher architectures are compatible, one can also distill intermediate activations for faster convergence and better transfer and generalization (Aguilar et al., 2020):

$$\mathcal{L}_{\text{reps}} = \sum_i ||r_i^s - r_i^t||^2, \tag{2}$$

where $r_i^s$ and $r_i^t$ are the corresponding transformer block output activations from student and teacher.

# 3 Robust binarization setup

In this section we first bring together some best practices and minor improvements which we have found helpful in simplifying previous work and building a strong baseline. Then we present a novel activation binarization scheme, which we will show to be critical to achieve good performance.

## 3.1 Two-set binarization scheme

In contrast to convolutional neural networks on images where activations exhibit comparable distributions, different activations in transformer blocks are performing different functionalities, and thus vary in their output distributions. In particular, these activations can be divided into two categories: the

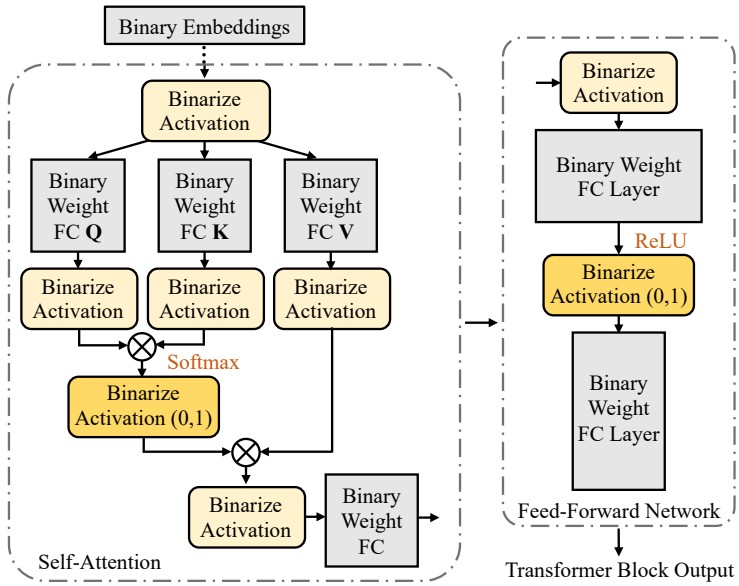

Figure 1: Overview of BiT. A transformer block contains the multi-head self-attention and feed-forward network. We binarize all the weights to {-1, 1} in the Embedding/Fully-Connected layers and binarize activations to {0, 1} for ReLU/Softmax outputs and to {-1, 1} for other layers.

activations after Softmax/ReLU layer that contains positive values only and the remaining activations with both positive and negative values (e.g., after matrix multiplication). If we denote by $\mathbf{X_R}$ the vector of activation values, then the two cases are $\mathbf{X_R}^i \in \mathbb{R}_+$ and $\mathbf{X_R}^i \in \mathbb{R}$ respectively.

For the former set, mapping to the binary levels $\{-1, 1\}$ would result in a severe distribution mismatch. Therefore we instead map non-negative activation layers to $\hat{\mathbf{X}}_{\mathbf{B}} \in \{0, 1\}^n$ and binarize activation layers with $\mathbf{X_R} \in \mathbb{R}^n$ to $\hat{\mathbf{X}}_{\mathbf{B}} \in \{-1, 1\}^n$, shown in Figure 1. A prior work BiBERT (Qin et al., 2021) also suggests binarizing attention to $\{0, 1\}$, but with $\mathrm{bool}$ function replacing $\mathrm{SoftMax}$, while we empirically find that simply binarizing attentions after $\mathrm{SoftMax}$ to $\{0, 1\}$ works better and binarizing $\mathrm{ReLU}$ output to $\{0, 1\}$ instead of $\{-1, 1\}$ brings further improvements. (See Section A.3 for details).

**Optimal scaling factor in two sets** Additionally, we apply a layer-wise scaling factor to binarized activations to reduce the binarization error, *i.e.*, $\mathbf{X_B} = \alpha \hat{\mathbf{X}}_{\mathbf{B}}$. The optimal values of $\alpha$ are different for the $\hat{\mathbf{X}}_{\mathbf{B}} \in \{0, 1\}^n$ and $\hat{\mathbf{X}}_{\mathbf{B}} \in \{-1, 1\}^n$ cases and can be calculated by minimizing the $l2$ error:

$$\mathcal{J}(\alpha) = ||\mathbf{X_R} - \alpha \hat{\mathbf{X}}_{\mathbf{B}}||^2$$
$$\alpha^* = \underset{\alpha \in \mathbb{R}_+}{\arg\min} \, \mathcal{J}(\alpha) \tag{3}$$

Following XNOR-Net (Rastegari et al., 2016), by expanding Eq. 3, we have

$$\mathcal{J}(\alpha) = \alpha^2 \hat{\mathbf{X}}_B^T \hat{\mathbf{X}}_{\mathbf{B}} - 2\alpha \mathbf{X_R}^T \hat{\mathbf{X}}_{\mathbf{B}} + \mathbf{X_R}^T \mathbf{X_R} \tag{4}$$

For the layers with $\mathbf{X_R} \in \mathbb{R}^n$ we follow the traditional methods of binarizing activations (Rastegari et al., 2016; Liu et al., 2018) by taking the sign of real-valued activations:

$$\hat{\mathbf{X}}_{\mathbf{B}}^i = \mathrm{Sign}(\mathbf{X_R}^i) = \begin{cases} -1, & \text{if } \mathbf{X_R}^i < 0 \\ +1, & \text{if } \mathbf{X_R}^i \geqslant 0 \end{cases} \tag{5}$$

In that case, $\hat{\mathbf{X}}_B^T \hat{\mathbf{X}}_{\mathbf{B}} = n_{\mathbf{X_R}}$, where $n_{\mathbf{X_R}}$ is number of elements in $\mathbf{X_R}$, and $\alpha^*$ can be solved as:

$$\alpha^* = \frac{\mathbf{X_R}^T \hat{\mathbf{X}}_{\mathbf{B}}}{n_{\mathbf{X_R}}} = \frac{||\mathbf{X_R}||_{l1}}{n_{\mathbf{X_R}}} \tag{6}$$

For the activations in attention layers or after the ReLU non-linearity layers with $\mathbf{X_R} \in \mathbb{R}_+^n$, we binarize the activations to $\hat{\mathbf{X}}_{\mathbf{B}} \in \{0, 1\}^n$ by rounding the real-valued activations:

$$\hat{\mathbf{X}}_{\mathbf{B}}^i = \lfloor \mathrm{Clip}(\mathbf{X_R}^i, 0, 1) \rceil = \begin{cases} 0, & \text{if } \mathbf{X_R}^i < 0.5 \\ 1, & \text{if } \mathbf{X_R}^i \geqslant 0.5 \end{cases} \tag{7}$$

In that case, $\hat{\mathbf{X}}_B^T \hat{\mathbf{X}}_\mathbf{B} = n_{\{\mathbf{X_R} \geqslant 0.5\}}$ where $n_{\{\mathbf{X_R} \geqslant 0.5\}}$ denotes the number of elements in $\mathbf{X_R}$ that are greater than or equal to $0.5$. Then $\alpha^*$ can be solved as:

$$\alpha^* = \frac{||\mathbf{X_R} \cdot \mathbf{1}_{\{\mathbf{X_R} \geqslant 0.5\}}||_{l1}}{n_{\{\mathbf{X_R} \geqslant 0.5\}}} \tag{8}$$

## 3.2   Best practices

We performed thorough experimentation and discovered the following modifications to be useful.

**Simplified knowledge distillation** Compared to previous BERT model binarization works (Bai et al., 2021; Qin et al., 2021) which also attempt to distill the attention scores, we provide analysis and experimental results (Section 5.3) showing that using only $\mathcal{L}_{\text{reps}}$ from transformer block outputs and $\mathcal{L}_{\text{logits}}$ is more effective while being simpler. We also forego the two-step distillation scheme of Bai et al. (2021) in favor of a single step, joint distillation, where our training loss is simply $\mathcal{L}_{\text{logits}} + \mathcal{L}_{\text{reps}}$.

**Mean subtraction in weight binarization** For weight binarization, centeralizing the real-valued weights to be zero-mean before binarization can increase the information carrying capacity of the binary weights. Thus, for weight binarization, we have: $\mathbf{W_B}^i = \frac{||\mathbf{W_R}||_{l1}}{n_{\mathbf{W_R}}} \text{Sign}(\mathbf{W_R}^i - \overline{\mathbf{W}}_\mathbf{R})$.

**Gradient clipping** Clipping gradients to 0 when $\mathbf{X_R}^i \notin [-1, 1]$ (or $\mathbf{X_R}^i \notin [0, 1]$ if $\hat{\mathbf{X}}_\mathbf{B} \in \{0, 1\}^n$) is a common technique for training binarized neural networks. However, we find that clipping weight gradients is harmful for optimization. Once a weight is outside of the clip range, the gradient is fixed to 0, preventing further learning. This is not so for activations, since the activation value changes for each input. As a result, we apply gradient clipping only to activations but not to weights.

**Non-linearity** We prefer ReLU activations whenever the output range is non-negative.

Combining these, we are able to build a strong baseline, which improves the accuracy by 9.6% over naively binarized transformers. Additionally, these techniques allow us to train a weight binarized transformer network in a single training step using knowledge distillation (*i.e.*, without resorting to weight splitting as in BinaryBert (Bai et al., 2021)) (See Section 5.3 for details).

## 3.3   Elastic binarization function

The fixed scaling and threshold derived previously works reasonably well, but might not be optimal since it ignores the distribution of the variable which is being binarized. Ideally, these parameters can be learned during training to minimize the target loss.

When using classical binarization methods, *i.e.*, $\hat{\mathbf{X}}_\mathbf{B}^i = \text{Sign}(\mathbf{X_R}^i)$, the binary output is independent of the scale of the real-valued input. However, in our case where $\hat{\mathbf{X}}_\mathbf{B}^i = \lfloor \text{Clip}(\mathbf{X_R}^i, 0, 1) \rceil$, this independence no longer holds. Learning the scaling and threshold parameters, and how to approximate the gradients precisely in the process becomes crucial for the final accuracy.

To handle this, inspired by the learnable threshold in ReActNet (Liu et al., 2020), we propose the elastic binarization function to learn both the scale $\alpha \in \mathbb{R}_+$ and the threshold $\beta \in \mathbb{R}$:

$$\mathbf{X_B}^i = \alpha \hat{\mathbf{X}}_\mathbf{B}^i = \alpha \lfloor \text{Clip}(\frac{\mathbf{X_R}^i - \beta}{\alpha}, 0, 1) \rceil \tag{9}$$

In the function, we initialize $\alpha$ with $\alpha^*$ in Sec. 3.1 and $\beta$ to be 0, and train it with gradients from the final loss. To back-propagate the gradients to $\alpha$ through the discretized binarization function, we follow the practice in Choi et al. (2018); Zhou et al. (2016); Esser et al. (2019) to use straight-through estimator (STE) (Bengio et al., 2013) to bypass the incoming gradients to the round function to be

---

**Algorithm 1** `BiT`: Multi-distillation algorithm

---

**Require:** $\mathcal{D}_{\text{train}}, \mathcal{D}_{\text{dev}}$                                                 ▷ Training Data
**Require:** $h_0$                                                      ▷ Full-precision Model
**Require:** $\mathbf{Q} = \{(b_w^1, b_a^1), \ldots, (b_w^k, b_a^k)\}$                                ▷ Quantization Schedule
1: $h_{\text{teacher}} \leftarrow h_0$
2: **for** $b_w^i, b_a^i$ in $\mathbf{Q}$ **do**
3:      $h_{\text{student}} \leftarrow \text{Quantize}(h_{\text{teacher}}, b_w^i, b_a^i)$
4:      $\text{KnowledgeDistill}(h_{\text{student}}, h_{\text{teacher}}, \mathcal{D}_{\text{train}}, \mathcal{D}_{\text{dev}})$
5:      $h_{\text{teacher}} \leftarrow h_{\text{student}}$
6: **end for**
7: **return** $h_{\text{student}}$

---

the outgoing gradients:

$$
\begin{aligned}
\frac{\partial \mathbf{X}_{\mathbf{B}}^i}{\partial \alpha} &= \hat{\mathbf{X}}_{\mathbf{B}}^i + \alpha \frac{\partial \hat{\mathbf{X}}_{\mathbf{B}}^i}{\partial \alpha} \\
&\overset{STE}{\approx} \hat{\mathbf{X}}_{\mathbf{B}}^i + \alpha \frac{\partial \text{Clip}(\frac{\mathbf{X}_{\mathbf{R}}^i - \beta}{\alpha}, 0, 1)}{\partial \alpha} \\
&= \begin{cases}
0, & \text{if } \mathbf{X}_{\mathbf{R}}^i < \beta \\
\frac{\beta - \mathbf{X}_{\mathbf{R}}^i}{\alpha}, & \text{if } \beta \leqslant \mathbf{X}_{\mathbf{R}}^i < \alpha/2 + \beta \\
1 - \frac{\mathbf{X}_{\mathbf{R}}^i - \beta}{\alpha}, & \text{if } \alpha/2 + \beta \leqslant \mathbf{X}_{\mathbf{R}}^i < \alpha + \beta \\
1, & \text{if } \mathbf{X}_{\mathbf{R}}^i \geqslant \alpha + \beta
\end{cases}
\end{aligned}
\tag{10}
$$

Then the gradients *w.r.t.* $\beta$ can be similarly calculates as:

$$
\frac{\partial \mathbf{X}_{\mathbf{B}}^i}{\partial \beta} \overset{STE}{\approx} \alpha \frac{\partial \text{Clip}(\frac{\mathbf{X}_{\mathbf{R}}^i - \beta}{\alpha}, 0, 1)}{\partial \beta} = \begin{cases} -1, & \text{if } \beta \leqslant \mathbf{X}_{\mathbf{R}}^i < \alpha + \beta \\ 0, & \text{otherwise} \end{cases}
\tag{11}
$$

For the layers that contain both positive and negative real-valued activations *i.e.*, $\mathbf{X}_{\mathbf{R}} \in \mathbb{R}^n$, the binarized values $\hat{\mathbf{X}}_{\mathbf{B}} \in \{-1, 1\}^n$ are indifferent to the scale inside the Sign function:

$$
\mathbf{X}_{\mathbf{B}}^i = \alpha \cdot \text{Sign}(\frac{\mathbf{X}_{\mathbf{R}}^i - \beta}{\alpha}) = \alpha \cdot \text{Sign}(\mathbf{X}_{\mathbf{R}}^i - \beta)
\tag{12}
$$

In that case, since the effect of scaling factor $\alpha$ inside the Sign function can be ignored, the gradient *w.r.t.* $\alpha$ can be simply calculated as:

$$
\frac{\partial \mathbf{X}_{\mathbf{B}}^i}{\partial \alpha} = \text{Sign}(\mathbf{X}_{\mathbf{R}}^i - \beta)
\tag{13}
$$

In our ablations (Section 5.3 and A.2) we show that using this simple elastic binarization function can bring a 15.7% accuracy boost over our strong baseline on the GLUE benchmark.

## 4   Multi-distilled binary transformer

Classical knowledge distillation (KD) (Hinton et al., 2015) trains the outputs (*i.e.*, logits) of a student network to be close to those of a teacher, which is typically larger and more complex. This approach is quite general, and can work with any student-teacher pair which conforms to the same output space. However, in practice, knowledge transfer happens faster and more effectively if the intermediate representations are also distilled (Aguilar et al., 2020). This approach has been found useful when distilling to student models with similar architecture (Sanh et al., 2019), and in particular for quantization (Bai et al., 2021; Kim et al., 2019).

Note that having a similar student-teacher pair is a requirement for distilling representations. While how similar they need to be is an open question, intuitively a teacher which is architecturally closer to the student should make transfer of internal representations easier. In the context of quantization, it is easy to see that lower precision students are progressively less similar to the full-precision teacher, which is one reason why binarization is difficult.

Table 1: Comparison of BERT quantization methods on the GLUE dev set. The E-W-A notation refers to the quantization level of embeddings, weights and activations. ‡ denotes distilling binary models using full-precision teacher without using multi-distill technique in Section 4. *Data augmentation is not needed for MNLI, QNLI, therefore results in the data augmentation section are identical to that without data augmentation for these datasets.

| Quant | #Bits | Size (MB) | FLOPs (G) | MNLI -m/mm | QQP | QNLI | SST-2 | CoLA | STS-B | MRPC | RTE | Avg. |
|---|---|---|---|---|---|---|---|---|---|---|---|---|
| BERT | 32-32-32 | 418 | 22.5 | 84.9/85.5 | 91.4 | 92.1 | 93.2 | 59.7 | 90.1 | 86.3 | 72.2 | 83.9 |
| *Without data augmentation* | | | | | | | | | | | | |
| Q-BERT | 2-8-8 | 43.0 | 6.5 | 76.6/77.0 | – | – | 84.6 | – | – | 68.3 | 52.7 | – |
| Q2BERT | 2-8-8 | 43.0 | 6.5 | 47.2/47.3 | 67.0 | 61.3 | 80.6 | 0 | 4.4 | 68.4 | 52.7 | 47.7 |
| TernaryBERT | 2-2-8 | 28.0 | 6.4 | 83.3/83.3 | 90.1 | – | – | 50.7 | – | 87.5 | 68.2 | – |
| BinaryBERT | 1-1-8 | 16.5 | 3.1 | 84.2/84.7 | 91.2 | 91.5 | 92.6 | 53.4 | 88.6 | 85.5 | 72.2 | 82.7 |
| BinaryBERT | 1-1-4 | 16.5 | 1.5 | 83.9/84.2 | 91.2 | 90.9 | 92.3 | 44.4 | 87.2 | 83.3 | 65.3 | 79.9 |
| BinaryBERT | 1-1-2 | 16.5 | 0.8 | 62.7/63.9 | 79.9 | 52.6 | 82.5 | 14.6 | 6.5 | 68.3 | 52.7 | 53.7 |
| BinaryBERT | 1-1-1 | 16.5 | 0.4 | 35.6/35.3 | 66.2 | 51.5 | 53.2 | 0 | 6.1 | 68.3 | 52.7 | 41.0 |
| BiBERT | 1-1-1 | 13.4 | 0.4 | 66.1/67.5 | 84.8 | 72.6 | 88.7 | 25.4 | 33.6 | 72.5 | 57.4 | 63.2 |
| BiT ‡ | 1-1-4 | 13.4 | 1.5 | 83.6/84.4 | 87.8 | 91.3 | 91.5 | 42.0 | 86.3 | 86.8 | 66.4 | 79.5 |
| BiT ‡ | 1-1-2 | 13.4 | 0.8 | 82.1/82.5 | 87.1 | 89.3 | 90.8 | 32.1 | 82.2 | 78.4 | 58.1 | 75.0 |
| BiT ‡ | 1-1-1 | 13.4 | 0.4 | 77.1/77.5 | 82.9 | 85.7 | 87.7 | 25.1 | 71.1 | 79.7 | 58.8 | 71.0 |
| BiT | **1-1-1** | **13.4** | **0.4** | **79.5/79.4** | **85.4** | **86.4** | **89.9** | **32.9** | **72.0** | **79.9** | **62.1** | **73.5** |
| *With data augmentation* | | | | | | | | | | | | |
| TernaryBERT | 2-2-8 | 28.0 | 6.4 | 83.3/83.3* | 90.1* | 90.0 | 92.9 | 47.8 | 84.3 | 82.6 | 68.4 | 80.3 |
| BinaryBERT | 1-1-8 | 16.5 | 3.1 | 84.2/84.7* | 91.2* | 91.6 | 93.2 | 55.5 | 89.2 | 86.0 | 74.0 | 83.3 |
| BinaryBERT | 1-1-4 | 16.5 | 1.5 | 83.9/84.2* | 91.2* | 91.4 | 93.7 | 53.3 | 88.6 | 86.0 | 71.5 | 82.6 |
| BinaryBERT | 1-1-2 | 16.5 | 0.8 | 62.7/63.9* | 79.9* | 51.0 | 89.6 | 33.0 | 11.4 | 71.0 | 55.9 | 57.6 |
| BinaryBERT | 1-1-1 | 16.5 | 0.4 | 35.6/35.3* | 66.2* | 66.1 | 78.3 | 7.3 | 22.1 | 69.3 | 57.7 | 48.7 |
| BiBERT | 1-1-1 | 13.4 | 0.4 | 66.1/67.5* | 84.8* | 76.0 | 90.9 | 37.8 | 56.7 | 78.8 | 61.0 | 68.8 |
| BiT ‡ | 1-1-2 | 13.4 | 0.8 | 82.1/82.5* | 87.1* | 88.8 | 92.5 | 43.2 | 86.3 | 90.4 | 72.9 | 80.4 |
| BiT ‡ | 1-1-1 | 13.4 | 0.4 | 77.1/77.5* | 82.9* | 85.0 | 91.5 | 32.0 | 84.1 | 88.0 | 67.5 | 76.0 |
| BiT | **1-1-1** | **13.4** | **0.4** | **79.5/79.4*** | **85.4*** | **86.5** | **92.3** | **38.2** | **84.2** | **88.0** | **69.7** | **78.0** |

This suggests a multi-step approach, where instead of directly distilling from a full-precision teacher to the desired quantization level, we first distill into a model with sufficient precision in order to preserve quality. This model can then be used as a teacher to distill into a further quantized student. This process can be repeated multiple times, while at each step ensuring that the teacher and student models are sufficiently similar, and the performance loss is limited. This multi-distillation approach is sketched in Algorithm 1.

The multi-step distillation follows a *quantization schedule*, $\mathbf{Q} = \{(b_w^1, b_a^1), (b_w^2, b_a^2), \ldots, (b_w^k, b_a^k)\}$ with $(b_w^1, b_a^1) > (b_w^2, b_a^2) > \ldots > (b_w^k, b_a^k)^3$. $(b_w^k, b_a^k)$ is the target quantization level, which is in our case binary for both weights and activations. In practice, we find that down to a quantization level of W1A2, we can distill models of reasonable accuracy in single shot, following the best practices outlined in Section 3.2 (See our 1-1-2 baseline results in Table 1). As a result, we follow a fixed quantization schedule, W32A32 → W1A2 → W1A1. This is not necessarily optimal, and how to efficiently find the best quantization schedule is an interesting open problem. We present our initial explorations towards this direction in Section 5.5.

Combining the elastic binary activations with multi-distillation we obtain `BiT`, the robustly binarized multi-distilled transformer. Note that `BiT` simultaneously ensures good initialization for the eventual (in our case binary) student model. Since the binary loss landscape is highly irregular, good initialization is critical to aid optimization. Previous work has proposed progressive distillation (Zhuang et al., 2018; Yang et al., 2019) to tackle this problem, wherein the student network is quantized at increasing severity as the training progresses. However, this method does not prevent the student network from drifting away from the teacher, which is always the full-precision model. We compare to progressive distillation in Section A.1.

## 5   Main results

We follow recent work (Bai et al., 2021; Qin et al., 2021) in adopting the experimental setting of Devlin et al. (2019), and use the pre-trained BERT-base as our full-precision baseline. We evaluate on GLUE (Wang et al., 2019), a varied set of language understanding tasks (see Section A.5 for a full list), as well as SQuAD (v1.1) (Rajpurkar et al., 2016), a popular machine reading comprehension dataset.

---

[3] $(a, b) > (c, d)$ if $a > c$ and $b \geq d$ or $a \geq c$ and $b > d$.

Table 2: Ablation study on the effects of each component on GLUE dataset without data augmentation.

| | Quant | MNLI$_{-m/mm}$ | QQP | QNLI | SST-2 | CoLA | STS-B | MRPC | RTE | Avg. |
|---|---|---|---|---|---|---|---|---|---|---|
| 1 | BERT$_{base}$ | 84.9/85.5 | 91.4 | 92.1 | 93.2 | 59.7 | 90.1 | 86.3 | 72.2 | 83.9 |
| 2 | BiBERT Baseline | 45.8/47.0 | 73.2 | 66.4 | 77.6 | 11.7 | 7.6 | 70.2 | 54.1 | 50.4 |
| 3 | BiBERT | 66.1/67.5 | 84.8 | 72.6 | 88.7 | 25.4 | 33.6 | 72.5 | 57.4 | 63.2 |
| 4 | BinaryBERT (Our implementation) | 36.2/35.9 | 59.6 | 52.4 | 65.6 | 9.3 | 19.8 | 69.9 | 52.7 | 45.7 |
| 5 | + Our simplied KD | 37.7/37.3 | 59.5 | 56.8 | 73.4 | 4.1 | 24.8 | 70.8 | 57.0 | 48.0 |
| 6 | + Our two-set binarization (Strong Baseline) | 57.4/59.1 | 68.3 | 64.7 | 81.0 | 18.2 | 24.7 | 71.8 | 56.7 | 55.3 |
| 7 | + Elastic binarization (BiT ‡) | 77.1/77.5 | 82.9 | 85.7 | 87.7 | 25.1 | 71.1 | 79.7 | 58.8 | 71.0 |
| 8 | + Multi-distillation (BiT) | **79.5/79.4** | **85.4** | **86.4** | **89.9** | **32.9** | **72.0** | **79.9** | **62.1** | **73.5** |

## 5.1 GLUE results

Our main results on the GLUE benchmarks are presented in Table 1. In the setting without data augmentation, where we only use the original training samples for knowledge distillation, we are able to reduce the gap to the full precision baseline by 49.8%, *i.e.*, from 20.7 in (Qin et al., 2021) to 10.4 points. We also see that our baseline models with elastic activation binarization already improve previous SoTA by large margins.

In the binary weight setting (4-bit activations), we can match or outperform Bai et al. (2021) without the need for pre-training half-width models and subsequently splitting weights. This result should make binary weight models much easier to implement and deploy.

We also set a new state of the art for binary weight 2-bit activation (W1A2) models, with only a 3.5 point degradation compared to the full-precision baseline (using data augmentation). While not as efficient as binary, 2-bit arithmetic can also be performed without multiplications, making it a good efficient alternative in applications where the performance cost of going to fully binary is significant.

### 5.1.1 Data augmentation

From Table 1, it can be observed that the datasets with small training sets still have a large gap from the full-precision baseline. As a result, we employ data augmentation heuristics (following the exact setup in Zhang et al. (2020)) on the datasets with small training sets (all except MNLI, QNLI) to take better advantage of our model's strong representational capability. This further reduces the quantization gap, with our models eventually trailing the full-precision model by only 5.9 points on average on the GLUE benchmark.

## 5.2 SQuAD results

We also evaluate on the popular machine reading comprehension (MRC) dataset from Rajpurkar et al. (2016). We compare to our own implementation of the MRC task on top of the BiBERT codebase, since SQuAD results are not reported in that work. We also show results using the BinaryBERT codebase, without using weight splitting. The results (Table 3) show that this task is significantly harder than most document classification benchmarks in GLUE, and previ-

Table 3: Comparison of BERT quantization methods on SQuADv1.1 dev set. Metrics are exact match and F1 score.

| Quant | #Bits | SQuADv1.1$_{EM/F1}$ |
|---|---|---|
| BERT$_{base}$ | 32-32-32 | 82.6/89.7 |
| BinaryBERT | 1-1-4 | 77.9/85.8 |
| BinaryBERT | 1-1-2 | 72.3/81.8 |
| BinaryBERT | 1-1-1 | 1.5/8.2 |
| BiBERT | 1-1-1 | 8.5/18.9 |
| BiT | **1-1-1** | **63.1/74.9** |

ous binarization methods fail to achieve any meaningful level of performance. BiT does much better, but still trails the 32-bit baseline by 14.8 points in F1. We conclude that despite the improvements we have demonstrated on the GLUE benchmark, binarizing transformer models accurately is far from a solved problem in general.

## 5.3 Ablations

We start from the basic binarization implementation from Bai et al. (2021), and add each of our contributions in sequence to get a better idea how each contributes to the performance. The results are shown in Table 2.

We start by removing attention distillation (Section 3.2), which results in a 2.3% improvement (row 5 vs. 4). Then switching to our two-set binarization (Section 3.1), which binarizes the attention scores

| | L 0 | L 1 | L 2 | L 3 | L 4 | L 5 | L 6 | L 7 | L 8 | L 9 | L 10 | L 11 |
|---|---|---|---|---|---|---|---|---|---|---|---|---|
| Query FC input | 0.91 | 0.85 | 0.86 | 0.65 | 0.64 | 0.47 | 0.54 | 0.44 | 0.64 | 0.67 | 0.68 | 0.59 |
| Key FC input | 0.97 | 0.93 | 0.87 | 0.6 | 0.56 | 0.44 | 0.51 | 0.48 | 0.64 | 0.64 | 0.7 | 0.65 |
| Value FC input | 0.48 | 0.42 | 0.21 | 0.25 | 0.32 | 0.03 | 0.11 | 0.04 | 0.24 | 0.39 | 0.73 | 0.83 |
| Query FC output | 1.49 | 1.68 | 1.93 | 1.83 | 1.74 | 1.45 | 1.58 | 1.33 | 1.54 | 1.55 | 1.64 | 1.64 |
| Key FC output | 1.46 | 1.58 | 1.66 | 1.62 | 1.46 | 1.17 | 1.32 | 1.19 | 1.41 | 1.35 | 1.42 | 1.42 |
| Value FC output | 1.01 | 1.17 | 0.56 | 0.77 | 0.97 | 0.1 | 0.32 | 0.1 | 0.55 | 0.71 | 0.94 | 1.08 |
| Attention | 0.01 | 0.01 | 0.04 | 0.01 | 0.01 | 0.1 | 0.03 | 0.12 | 0.01 | 0.01 | 0.01 | 0.01 |
| Self-Attention FC input | 0.01 | 0.03 | 0.02 | 0.02 | 0.02 | 0.02 | 0.02 | 0.02 | 0.03 | 0.03 | 0.03 | 0.04 |
| FFN 1st FC input | 0.05 | 0.06 | 0.06 | 0.06 | 0.04 | 0.03 | 0.03 | 0.03 | 0.03 | 0.77 | 0.8 | 0.86 |
| FFN 2nd FC input | 0.22 | 0.23 | 0.2 | 0.14 | 0.12 | 0.11 | 0.11 | 0.12 | 0.11 | 0.04 | 0.05 | 0.04 |

Figure 2: The optimized scaling factor in `BiT`

differently than the feed-forward activations, which gives an additional 7.3% boost (row 6). This results in a much stronger baseline than what was used in prior works (row 2).

Moving from fixed to elastic binarization (Section 3.3) proves hugely important, pushing the average accuracy to 71.0% (row 7) from only 55.3% (row 6). Note that this model already outperforms the current state-of-the-art (row 3) by 7.8% points. Finally, we add multi-step distillation (Section 4), which adds another 2.5 points, reaching the final accuracy of 73.5% on the GLUE benchmark.

### 5.4 Learned parameter visualization

We visualize the optimized $\alpha$ in the final `BiT` model. As we can see from Figure 2, the values of the $\alpha$ parameters vary significantly from layer to layer, and have apparent patterns according to layer characteristics. For example, the attention layers need to distribute the attention to different entries, thus the scaling factor for the attentions are learned to be small, while the scaling factors for the query and key outputs are usually larger. Note that the biggest $\alpha$ value is $200\times$ of the smallest $\alpha$, suggesting the importance of learning $\alpha$ dynamically.

### 5.5 Exploring multi-distillation paths

So far we have only considered the fixed quantization schedule, W32A32 $\rightarrow$ W1A2 $\rightarrow$ W1A1. This is motivated by early experiments showing that one-step distillation to W1A2 works reasonably well. We explored other optimal schedules, such as distilling to W1A8 resulted in a higher accuracy model, thus a better teacher to distill down to the eventual W1A1 student. This suggests a trade-off between the quality of the intermediate model, vs. the closeness to the target quantization level.

Figure 3, illustrates this trade-off. We can see that two-step distillation improves over one-step in every case. While higher precision intermediate models are better as expected, it is better to use a lower precision teacher in the last step since it makes the learning task for the binary student model easier. The closeness to the target quantization is favored despite the lower accuracy of teacher model.

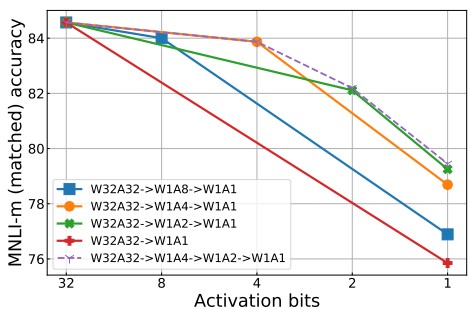

Figure 3: MNLI-m accuracy on various distillation paths. Each curve represents the sequence of models on a particular quantization schedule.

It is of course possible, though more cumbersome, to perform more than two distillation steps. We also experiment with a three-step schedule, W32A32 $\rightarrow$ W1A4 $\rightarrow$ W1A2 $\rightarrow$ W1A1, which is plotted in the same figure (dashed line). We find that this particular 3-step schedule does not improve over the 2-step schedule W32A32 $\rightarrow$ W1A2 $\rightarrow$ W1A1. While this result does not preclude existence of other more optimal schedules, we hypothesize that this is unlikely.

## 6 Related work

**Convolutional neural network quantization** Neural network quantization is a practical tool for compressing model size and reduce storage (Hubara et al., 2017). Quantization for convolutional

neural networks has been studied both in the uniform quantization (Choi et al., 2018; Zhou et al., 2016; Gong et al., 2019) and non-uniform quantization (Zhang et al., 2018; Miyashita et al., 2016; Li et al., 2020) settings. The quantization level has been progressively increased, from 8-bit (Wang et al., 2018; Zhu et al., 2020) to 4-bit (Jung et al., 2019; Esser et al., 2019; Liu et al., 2022) and finally to the extreme 1-bit case (Courbariaux et al., 2016; Rastegari et al., 2016; Liu et al., 2018; Martinez et al., 2020).

**Transformer quantization** Compared to the CNNs, transformers with attention layers are naturally more challenging to quantize (Bondarenko et al., 2021). Previous research mainly focused on 8-bit quantization (Zafrir et al., 2019; Fan et al., 2020) or 4-bit quantization (Shen et al., 2020; Zadeh et al., 2020). Extremely low-bit quantization for transformers has only been attempted very recently. TernaryBERT (Zhang et al., 2020) proposed to ternarize the full-precision weights of a fine-tuned BERT model. As a follow-up to TernaryBERT, weight binarization was proposed in Bai et al. (2021). Here, the network is trained by first training a ternary half-sized model, which is used as initialization. Then a weight-splitting step results in a full-sized binarized model, which is further fine-tuned in a subsequent distillation step. Binarizing both weight and activations in a transformer has proved to be challenging. BiBERT (Qin et al., 2021) made the first attempt in this direction with limited success. Their model performed 20% worse than a real-valued baseline on the GLUE benchmark (Wang et al., 2019), which even underperforms the original LSTM baselines.

# 7 Conclusion

Large pre-trained transformers have transformed NLP and are positioned to serve as the backbone for all AI models. In this work, we presented the first successful demonstration of a fully binary pre-trained transformer model. While our approach is general and can be applied to any transformer, we have limited our evaluation to BERT-based models on the GLUE and SQuAD benchmarks. It remains to be seen how our conclusions will hold when applied to the wide variety of pre-trained transformer models which have gained popularity in recent years, from small mobile models, to gigantic ones with hundreds of billions of parameters. It will also be interesting to see the performance of the approach on different domains (such as image and speech processing) and tasks (such as text and image generation). Demonstrating the generality of this approach in a wider setting should significantly widen its impact, therefore we identify this as an important future direction.

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
