# `BiT`: Robustly Binarized Multi-distilled Transformer

## A  Appendix

### A.1  `BiT` vs. progressive distillation

Table 1: `BiT` vs. progressive distillation on selected GLUE tasks. Methods differ in the teacher model used and the model from which the student weights are initialized.

| Method | Teacher | Initialization | MNLI$_{\text{-m/mm}}$ | QQP | QNLI | SST-2 | CoLA | STS-B | MRPC | RTE | Avg. |
|---|---|---|---|---|---|---|---|---|---|---|---|
| BiBERT Distillation | 32-32-32 | 32-32-32 | 77.0/77.2 | 83.1 | 84.1 | 89.7 | 31.3 | 60.1 | 75.5 | 56.7 | 69.7 |
| Progressive | 32-32-32 | 1-1-2 | 78.9/78.9 | 85.0 | 86.4 | 89.6 | 30.5 | **75.1** | **81.1** | 60.6 | 73.4 |
| BiT | **1-1-2** | **1-1-2** | **79.5/79.4** | **85.4** | **86.4** | **89.9** | **32.9** | 72.0 | 79.9 | **62.1** | **73.5** |

Previous work has also recognized the importance of good initialization for binary model training, and proposed to perform distillation while progressively quantizing the student model (Zhuang et al., 2018; Yang et al., 2019). Progressive distillation ensures a good initialization for the student model at each step. However, in this approach the teacher model is fixed to the full precision model, which does not address the problem of teacher-student gap. In Table 1 we compare `BiT` to a comparable implementation of progressive distillation, using the same quantization schedule, W32A32 → W1A2 → W1A1, as ours. We keep the teacher model fixed, while re-initializing the student model from the latest quantized version at each step. We see that using a quantized teacher model is helpful, especially in the high-data regime. However, our method can lag behind progressive distillation for small datasets such as STS-B and MRPC.

### A.2  Elastic binarization function vs. ReActNet learnable bias

Table 2: Elastic binarization function vs. ReActNet (Liu et al., 2020) learnable bias on GLUE tasks.

| Method | MNLI$_{\text{-m/mm}}$ | QQP | QNLI | SST-2 | CoLA | STS-B | MRPC | RTE | Avg. |
|---|---|---|---|---|---|---|---|---|---|
| Our two-set binarization (Strong Baseline) | 57.4/59.1 | 68.3 | 64.7 | 81.0 | 18.2 | 24.7 | 71.8 | 56.7 | 55.3 |
| + learnable scale | 76.5/76.8 | 82.7 | 85.1 | 88.1 | 26.6 | 62.3 | 74.3 | 58.1 | 69.2 |
| + learnable scale and bias (BiT ‡) | 77.1/77.5 | 82.9 | 85.7 | 87.7 | 25.1 | 71.1 | 79.7 | 58.8 | 71.0 |

Inspired by the learnable bias proposed in ReActNet (Liu et al., 2020), we further propose elastic binarization function to learn both learnable scaling factors and learnable bias. We find this learnable scaling factor critical for the final performance. As shown in table 2, the proposed learnable scaling factor brings 13.9% accuracy improvement, and further adding learnable bias boosts the accuracy by 1.8%.

### A.3  Two-set binarization scheme vs. Bi-Attention

In contrast to Bi-Attention proposed in BiBERT (Qin et al., 2021) that removes $\mathrm{SoftMax}$ and binarizes the attention to $\{0, 1\}$ with $\mathrm{bool}$ function, our two-set binarization scheme finds that keeping $\mathrm{SoftMax}$ in attention computation and also binarizing the positive output of $\mathrm{ReLU}$ layer to $\{0, 1\}$ works better. We conduct meticulous experiments to compare these choices. In Table 3, we show that, compared to removing $\mathrm{SoftMax}$ as Bi-Attention suggested, simply binarizing the activations after $\mathrm{SoftMax}$ layer

Table 3: Two-set binarization scheme vs. Bi-Attention (Qin et al., 2021) on GLUE tasks. Methods differ in whether using $\mathrm{SoftMax}$ in attention and whether binarizing the $\mathrm{ReLU}$ output to $\{0, 1\}$.

| Method | Attention | ReLU output | MNLI$_{\text{-m/mm}}$ | QQP | QNLI | SST-2 | CoLA | STS-B | MRPC | RTE | Avg. |
|---|---|---|---|---|---|---|---|---|---|---|---|
| Bi-Attention (w/o Softmax) | $\{0, 1\}$ | $\{-1, 1\}$ | 48.1/50.0 | 60.1 | 60.6 | 78.8 | 14.0 | 22.3 | 68.4 | 58.1 | 51.3 |
| Binarize attention to $\{0, 1\}$ (w/ Softmax) | $\{0, 1\}$ | $\{-1, 1\}$ | 51.9/52.6 | 76.2 | 60.5 | 79.6 | 11.6 | 18.1 | 70.6 | 55.6 | 53.0 |
| Two-set binarization scheme | $\{0, 1\}$ | $\{0, 1\}$ | 57.4/59.1 | 68.3 | 64.7 | 81.0 | 18.2 | 24.7 | 71.8 | 56.7 | 55.3 |

to $\{0, 1\}$ even produces 1.7% better accuracy. Furthermore, binarizing the $\mathrm{ReLU}$ layer output to $\{0, 1\}$ instead of $\{-1, 1\}$ helps the binary network match real-valued distributions and further brings 2.3% accuracy improvement.

## A.4   Binary convolution implementation for two-set binarization scheme

The binary convolution between the weights and activations that are both binarized to $\{-1, 1\}$ (i.e. $\mathbf{A_B} \in \{-1, 1\}$, $\mathbf{W_B} \in \{-1, 1\}$) can be implemented by the bitwise $\mathrm{xnor}$ operation followed by a $\mathrm{popcnt}$ operation (Rastegari et al., 2016; Liu et al., 2018):

$$\mathbf{A_B} \cdot \mathbf{W_B} = \mathrm{popcnt}(\mathrm{xnor}(\mathbf{A_B}, \mathbf{W_B})) \tag{1}$$

For the case where activations are binarized to $\{0, 1\}$ in two-set binarization scheme, the binary activation $\mathbf{A_B} \in \{0, 1\}$ can be represented with $\mathbf{A'_B} \in \{-1, 1\}$ through a simple linear mapping: $\mathbf{A_B} = \frac{\mathbf{A'_B}+1}{2}$. Thus the matrix computation between binary weights ($\mathbf{W_B} \in \{-1, 1\}$) and binary activations ($\mathbf{A_B} \in \{0, 1\}$) can be converted to the operations between $\mathbf{W_B} \in \{-1, 1\}$ and $\mathbf{A'_B} \in \{-1, 1\}$ as:

$$\mathbf{A_B} \cdot \mathbf{W_B} = (\frac{\mathbf{A'_B} + 1}{2}) \cdot \mathbf{W_B} = \frac{1}{2}(\mathrm{popcnt}(\mathrm{xnor}(\mathbf{A'_B}, \mathbf{W_B})) + \sum_i \mathbf{W_{B_i}}) \tag{2}$$

Here the $\sum_i \mathbf{W_{B_i}}$ is summing up the values in $\mathbf{W_B}$, which can be pre-computed and stored as bias. Thus in the two-set binarization scheme where activations are binarized to $\{0, 1\}$, the binary convolution can still be implemented with the general binary convolution in E.q. 1 at no additional complexity cost.

## A.5   Evaluation benchmarks

### A.5.1   GLUE

The GLUE benchmark (Wang et al., 2019) includes the following datasets:

**MNLI**   Multi-Genre Natural Language Inference is an entailment classification task (Williams et al., 2018). The goal is to predict whether a given sentence *entails*, *contradicts*, or is *neutral* with respect to another.

**QQP**   Quora Question Pairs is a paraphrase detection task. The goal is to classify whether two given questions have the same meaning. The questions were sourced from the Quora question answering website (Chen et al., 2018).

**QNLI**   Question Natural Language Inference (Wang et al., 2019) is a binary classification task which is derived from the Stanford Question Answering Dataset (Rajpurkar et al., 2016). The task is to predict whether a sentence contains the answer to a given question.

**SST-2**   The Stanford Sentiment Treebank is a binary sentiment classification task, with content taken from movie reviews (Socher et al., 2013).

**CoLA**   The Corpus of Linguistic Acceptability is a corpus of English sentences, each with a binary label denoting whether the sentence is linguistically acceptable (Warstadt et al., 2019).

**STS-B**   The Semantic Textual Similarity Benchmark is a sentence pair classification task. The goal is to predict how similar the two sentences are in meaning, with scores ranging from 1 to 5 (Cer et al., 2017).

**MRPC**  Microsoft Research Paraphrase Corpus is another sentence pair paraphrase detection task similar to QQP. The sentence pairs are sourced from online news sources (Dolan & Brockett, 2005).

**RTE**  Recognizing Textual Entailment is a small natural language inference dataset similar to MNLI in content (Bentivogli et al., 2009).

### A.5.2  SQuAD

The SQuAD benchmark (Rajpurkar et al., 2016), *i.e.*, Stanford Question Answering Dataset, is a reading comprehension dataset, consisting of questions on a set of Wikipedia articles, where the answer to each question is a segment of text from the corresponding passage, or the question might be unanswerable.

### A.6  Technical details

For each experiment, we sweep the learning rate in {1e-4, 2e-4, 5e-4} and the batch size in {8, 16} for QNLI, SST-2, CoLA, STS-B, MRPC, RTE, and {16, 32} for MNLI, QQP as well as SQuAD, and choose the settings with the highest accuracy on the validation set. We use the same number of training epochs as BiBERT (Qin et al., 2021), *i.e.*, 50 for CoLA, 20 for MRPC, STS-B and RTE, 10 for SST-2 and QNLI, 5 for MNLI and QQP. We adopt the Adam optimizer with weight decay 0.01 and use 0.1 warmup ratio with linear learning rate decay.

Our full precision checkpoints are taken from `https://textattack.readthedocs.io/en/latest/3recipes/models.html#bert-base-uncased`.