# OpenReview forum: "BiT: Robustly Binarized Multi-distilled Transformer"
_NeurIPS.cc/2022/Conference — NeurIPS 2022 Accept_

### Official Review · Reviewer_Yoaz · 2022-07-11

**Rating:** 7
**Confidence:** 3
**Soundness:** 3 good
**Presentation:** 3 good
**Contribution:** 3 good

**Summary:**

The authors propose novel binarisation techniques for transformer models based on knowledge distillation. The goal is to achieve competitive performance for an efficient student transformer model.  The main contributions are: i) binarisation technique that improves knowledge distillation in transformer models, ii) multi-step technique to distil binarised models to improve intermediate performance. The study shows that the student model achieves competitive performance compared to a standard BERT on the GLUE benchmark and question answering.

**Questions:**

Questions to the Authors


Please address the following questions during the rebuttal:


- Could you elaborate on the hyper-parameter selection for the teacher-student.

- A possible extra contribution is to perform multiple  random runs and report variance. However, how expensive could this exercise become?

- Please speculate if by using other pre-training objectives in the LM task (e.g. next sentence) would it change any finding or results?

**Limitations:**

The authors have addressed limitations of the proposed approach.

**Strengths And Weaknesses:**

Strengths


- Clear description of background knowledge and related work needed to understand the proposed approach.

- Clear description of the proposed approach.

- Best practices for model replicability.

- The authors perform a comprehensive comparison with related work on the GLUE benchmark.

- The findings show that an efficient transformer model has competitive performance compared to previous SOTA.


Weaknesses


- It is not clear how the parameter initialization  and selection of hyper-parameters could affect the model performance.

---

> ### Author Response · Authors · 2022-08-02
> **Author Response**
>
> Thanks for the supportive comments and detailed review.
>
> **Q1**: Elaborate on the hyper-parameter selection for the teacher-student.
>
> **A1**: For knowledge distillation (KD) hyperparameters, we set the KD temperature ($\mathcal{T}$) to 1 in the KD-loss: $-\sum \rm{softmax}(\frac{logits_{teacher}}{\mathcal{T}})\rm{logsoftmax}(\frac{logits_{student}}{\mathcal{T}})$ in all the distillation experiments. We also tried using $\mathcal{T}=3$, which is another common setting for distillation, but did not find much difference in the final accuracy.
> For the training batch size and learning rate. We choose the learning rate in {1e-4, 2e-4, 5e-4} and batch size in {16, 32} from the dev set (as mentioned in A.6), following the common practice in fine-tuning BERT models. Different hyperparameter settings yield ~1.5% accuracy difference.
>
> **Q2**: A possible extra contribution is to perform multiple random runs and report variance. However, how expensive could this exercise become?
>
> **A2**: Thanks for the question. We tried running the same experiment for two trials, but that produced identical accuracy across all GLUE datasets. We deem that is because the student networks are all initialized from the pre-trained BERT model instead of random initialization, which overcomes the potential variance brought by the randomness.
>
> **Q3**: Please speculate if by using other pre-training objectives in the LM task (e.g. next sentence) would it change any finding or results?
>
> **A3**: We discussed this interesting topic and think that the pre-training objectives will affect the final absolute accuracy, but will not influence too much on the relative accuracy gap between the real-valued model and the binarized model. This is because the accuracy gap depends mostly on three factors: (1) model capability (2) compression algorithm (3) the fine-tuning dataset and scheme. Since these three factors remain unchanged, we think the accuracy gap should be relatively consistent across different pre-training objectives.

---

> > ### Comment · Reviewer_Yoaz · 2022-08-08
> > **Rebuttal**
> >
> > Thank you, the authors have addressed my questions during rebuttal.

---

### Official Review · Reviewer_2rah · 2022-07-12

**Rating:** 5
**Confidence:** 3
**Soundness:** 2 fair
**Presentation:** 3 good
**Contribution:** 2 fair

**Summary:**

This paper proposes several changes to the design of the binary BERT model. The main proposal consists of three parts: 1) use different quantization schemes based on the output distribution of a layer (e.g., activations are quantized to 0 and 1 for multi-head self attention layers); 2) adopt an elastic binarization function which allows re-scale and shift; 3) use a multi-step distillation scheme joint with a multi-step quantization schedule. The resulting BiT model demonstrates a strong performance on downstream tasks in GLUE and SQuAD datasets.

**Questions:**

I would like to hear a response on the relationship to related work and also an analysis on the hardware complexity/cost of the BiT model. In addition, the authors could clarify the training cost of the proposed multi-step distillation scheme.

**Strengths And Weaknesses:**

Strengths: This paper proposes several simple yet effective changes to improve the quality of Binarized BERT model significantly. The proposed techniques are sound. The writing of the paper is clear and easy to follow. The authors also provide enough empirical evaluation of the proposed BiT models.

Weaknesses: My biggest concern with this paper is that the authors do not clearly discuss the relationship to other recent works. In the absence of a clear discussion of the relationship to other related work, I am not convinced that the contributions in this work are new and original.

This paper does cite a large number of relevant works. However, the relationship between the proposals in this work and related work remains unclear. First, the proposal to use different quantization schemes depending on the output distribution of the layers is quite similar to Bi-Attention proposed in BiBERT [1]. Bi-Attention suggests using a bool function (Equation 11 in the BiBERT paper) rather than a sign function to binarize the output of a multi-head self-attention layer. Second, the elastic binarization function looks akin to the activation function proposed in Reactnet [2]. Reactnet and some follow-up work have already shown that introducing additional scaling and shifting capabilities in the activation function can help improve the quality of the binarized CNN. The authors do not point out the similarity with Reactnet when discussing the elastic binarization function, which could be misleading.

In addition, I think the paper also needs to discuss the new hardware required for the BiT model, which is different from traditional binary neural networks that require only simple XNOR gates to implement the computational engine/processing element. The computational engine for BiT models needs to handle computation between 1 and -1 and 0 and 1. I suggest that the authors should at least compare the required hardware with traditional binary and ternary neural networks. Otherwise, it would be difficult to justify not using a ternary neural network.

[1] BiBERT: Accurate Fully Binarized BERT, ICLR'22
[2] Reactnet: Towards precise binary neural network with generalized activation functions, ECCV'20

---

> ### Author Response · Authors · 2022-08-02
> **Author Response**
>
> Thanks for the constructive feedback and detailed comments. We have clarified the differences from recent work and provided analysis of hardware complexity in the paper and response, which should address the concern. We respectfully ask the reviewer to reconsider the rating, since this review is quite positive otherwise.
>
> **Q1**: The relationship to Bi-Attention.
>
> **A1**: We agree that the proposed two-set binarization and Bi-Attention both use {0, 1} to represent the binarized attention. However, Bi-Attention replaces softmax with the bool function (Eq.11 in BiBERT), while we find that simply binarizing the softmax output to {0, 1} works better (1.7% higher on GLUE dataset). Moreover, we discover that binarizing the ReLU output to {0, 1} is also important, which further brings 2.3% accuracy improvement.
>
> We did these comparison experiments but did not include them in the paper due to the page limit. The results are as follows:
>
> |Method|Attention|ReLU output|MNLI$_{-m/mm}$|QQP|QNLI|SST-2|CoLA|STS-B|MRPC|RTE|**Avg.**|
> |-|-|-|-|-|-|-|-|-|-|-|-|
> |Bi-Attention with bool function, no softmax|{0, 1}|{-1, 1}|48.1/50.0|60.1|60.6|78.8|14.0|22.3|68.4|58.1|**51.3**|
> |Binarize attention to {0, 1}, keep softmax|{0, 1}|{-1, 1}|51.9/52.6|76.2|60.5|79.6|11.6|18.1|70.6|55.6|**53.0**|
> |Two-set binarization (Table 2 row 6 in paper)|{0, 1}|{0, 1}|57.4/59.1|68.3|64.7|81.0|18.2|24.7|71.8|56.7|**55.3**|
>
> We have added the discussion with BiBERT in Sec. 3.1 and included the above ablation study in Appendix (Sec. A.3) to give readers a more comprehensive understanding.
>
> **Q2**: Elastic binarization vs. ReActNet.
>
> **A2**: We agree that both ReActNet and the proposed elastic binarization use learnable bias. The difference is that elastic binarization proposes a novel learnable scaling factor that contributes to the major accuracy improvement. In our experiments, the learnable scaling factor brings 13.9% improvement on GLUE compared to 1.8% by learnable bias, shown in the table below:
>
> |Method|MNLI$_{-m/mm}$|QQP|QNLI|SST-2|CoLA|STS-B|MRPC|RTE|**Avg.**|
> |-|-|-|-|-|-|-|-|-|-|
> |Our two-set binarization (Strong Baseline)|57.4/59.1|68.3|64.7|81.0|18.2|24.7|71.8|56.7|**55.3**|
> |+ learnable scale $\alpha$|76.5/76.8|82.7|85.1|88.1|26.6|62.3|74.3|58.1|**69.2**|
> |+ learnable scale $\alpha$ and bias $\beta$ (BiT$\ddagger$)|77.1/77.5|82.9|85.7|87.7|25.1|71.1|79.7|58.8|**71.0**|
>
> Note that the proposed learnable scale is especially important for the case $A_B\in$ {0, 1}, because in that case the scale of real-valued activations matters for the binary outputs, i.e., $\lfloor Clip(\alpha X,0,1)\rceil\not=\lfloor Clip(X,0,1)\rceil$. However, this scenario is seldom studied before since in most previous works, $A_B\in$ {-1, 1}, for which scaling real-valued activations makes no difference, i.e., $Sign(\alpha X)=Sign(X)$. We have added these discussions and experiments to Appendix (Sec. A.2) to make the paper clearer.
>
> **Q3**: Hardware implementation complexity of BiT model.
>
> **A3**: In BiT, weights are binarized to {-1, 1} and the two-set binarization scheme only applies to activations. Binarizing activations to {0, 1} requires no additional hardware adjustment compared to binarizing to {-1, 1}. Because we can represent the binary activation $A_B\in$ {0, 1} with $A’_B\in$ {-1, 1} through a simple linear mapping:
> $$A_B=\frac{A’_B+1}{2}$$
> Thus the matrix multiplication between binary weights ($W_B\in$ {-1, 1}) and binary activations ($A_B\in$ {0, 1}) can be converted to the operations between $W_B\in$ {-1, 1} and $A'_B\in$ {-1, 1} as:
> $$W_B^T A_B=W_B^T (\frac{A'_B+1}{2})=\frac{1}{2}(popcnt(xnor(W_B,A'_B)+\Sigma W_B)$$
> Here $\Sigma W_B$ is summing up values in $W_B$, which can be pre-computed and stored as bias. Therefore, implementing BiT incurs no additional complexity.
>
> **Q4**: The training cost of multi-step distillation.
>
> **A4**: For the multi-step distillation, we use the same number of training epochs as BiBERT in each step, i.e., 50 for CoLA, 20 for MRPC, STS-B and RTE, 10 for SST-2 and QNLI, 5 for MNLI and QQP, as mentioned in Sec. A.6.
>
> Multi-step distillation does incur more training time. However, we have to stress that the better performance of multi-step distillation does not come from more training. We compared two-step distillation (BiT) with using 2$\times$ training epochs for single-step distillation (BiT$\ddagger$). It turns out that simply doubling the training time in single-step distillation yields no accuracy improvement, suggesting that the original recipe is already sufficient for training the 1-bit model to fully converge. In contrast, multi-step distillation further improves the accuracy by closing the gap between student and teacher models.
>
> Since the BiT$\ddagger$ with single-step distillation already exceeded the state-of-the-art by 7.8%, the multi-step distillation can also be regarded as a plus for scenarios with greater training time tolerance but higher requirement on inference accuracy.

---

> > ### Comment · Reviewer_2rah · 2022-08-08
> > **Post-rebuttal**
> >
> > Thank the authors for the response. These clarifications are important to understand the contribution of this work. I now see that small changes to existing proposals can lead to significant improvements in quality. I will raise my rating to borderline accept.

---

### Official Review · Reviewer_4xBa · 2022-07-19

**Rating:** 7
**Confidence:** 4
**Soundness:** 4 excellent
**Presentation:** 4 excellent
**Contribution:** 3 good

**Summary:**

This paper addresses the task of quantizing transformers to very low precision (1/2 bit(s)). Quantizing neural networks can be essential for model size reduction (especially for mobile devices), but also for the speed of evaluation of hardware without floating point accelerators. In addition, binary activations/computations open new possibilities for high-performing low-power, special purpose hardware for evaluating neural networks with drastically improved parallelism and power consumption.

While these considerations and motivations are not new, the paper gives a concise summary and references to prior efforts, especially in the context of convolutional networks, that is clearly referenced in this work.

Here this work addresses a set of cumulative, technical improvements for successfully quantizing transformer networks for very low precision, including normalization, separate quantization for non-negative activations and exploring distillation paths over medium precision (eg. 8 bits) quantization. Using a combination of such methods, this paper manages to reduce gap in BLUE score to the full-precision model by ~3X percentage points. Also the paper gives thorough ablation analysis for the effect of the employed methods.

**Questions:**

- It would be interesting to see whether a -1, 0, 1 quantization rather than the proposed -1, 1 would yield significant improvements at a moderate cost. Has this been attempted?
- How does quantization affect the quality of vision transformers?

**Limitations:**

- While every improvement has some potential to increase the risk of powerful technologies. It is unclear/unlikely to me that this technology would have any adverse affects beyond the generic issues of making high-impact technologies easier/cheaper to deploy by bad actors and that approximate inference might become less reliable than high-quality models. These are a general concerns regarding any performance improvements and performance trade-off. In this sense, I don't think that this approach has any non-generic risks that should have been considered in particular by the authors.

**Strengths And Weaknesses:**

Originality: Moderate. The motivation for extreme quantization of neural networks is not new and has been proposed many times over the past few years. These include model size reduction and the development of new special purpose hardware that could reduce the power consumption and latency of inference of large deep learning models. In fact the feasibility of extreme quantization has been pioneered for convolutional networks with high success, but with transformers becoming the main workhorse for most deep learning applications, it has become an important question how to transfer the above results over to them.

While there has been an initial set of work of binarizing BERT and other transformer-based models, typically those results resulted in a huge drop of language-modeling quality.

This work does not propose one single solution to those issues, but a collection of relatively common-sense solutions, the combination of which has a large effect on quality of the resulting low-precision model. These ideas include:
- Normalizing activations before binarization to ensure that low-precision activations are maximally informative.
- Improved gradient clipping for training.
- Specialized quantization for non-negative and general activations.
- Attention quantization
- Intermediate-layer distillation (employed in prior work)
- Multi-step distillation path over medium-precision models.

Quality: High. The paper has one very clear goal, a very well-defined experimental setup. Thorough experiments and ablation analysis that seem consistent. The work presents a clear analysis of the importance of all the employed methods. While most of the ideas/methods are not very novel in isolation, this work gives a clear, well-tested recipe for the extreme quantization of transformers the quality of which is significantly beyond whatever was reported earlier. While the paper omits speculating about the potential gains by special-purpose hardware, this is not a real issue, as the implications are relatively clear and these results give a clear motivation to further work in that direction.

Clarity: High. The paper is well motivated. References to prior work is quite extensive, although references to very new, concurrent works might (that has not been peer-reviewed yet), but this does not affect the final conclusion.  The paper presents a lot of experimental evidence in concise tables that back up the intuition behind the decision and highlight the significance and motivation for all of the decisions made for this work.

Significance: High. While the employed methods are relatively well known, the fact that transformers can be quantized to 1-bit precision is extremely important and this work gives a clear, well-documented measurement point and clear, well-tested recipes. This is a valuable baseline for future work, also increases the motivation for building special-purpose hardware for extremely low-precision neural networks.

---

> ### Author Response · Authors · 2022-08-02
> **Author Response**
>
> Thank you very much for the detailed comments and the precise summarization of our work.
>
> **Q1**: It would be interesting to see whether a -1, 0, 1 quantization rather than the proposed -1, 1 would yield significant improvements at a moderate cost. Has this been attempted?
>
> **A1**: Yes, we did an experiment on ternary quantization, the result is as follows:
>
> | Method | MNLI$_{-m/mm}$ | QQP | QNLI | SST-2 | CoLA | STS-B | MRPC | RTE | Avg.|
> | --- | --- | --- | --- | --- | --- | --- | --- | --- | --- |
> | Ternary-W-Ternary-A | 80.9/81.3 | 86.6 | 87.6 | 89.9 | 29.1 | 81.2 | 76.0 | 59.6 | **73.9** |
> | W1A1 (BiT $\ddagger$ )| 77.1/77.5 | 82.9 | 85.7 | 87.7 | 25.1 | 71.1 | 79.7 | 58.8 | **71.0** |
>
> Compared to the binary model distilled from the full-precision BERT model, the ternary model with the same settings achieves 2.9% higher accuracy on the GLUE dataset.
>
> For the cost of ternary models, to our knowledge, there are two methods to implement ternary computation: (1) Treating the ‘0’ value as sparsity, which has the potential of further saving the memory. But it requires a certain level of sparsity and specially designed kernel/implementation to enjoy the actual memory saving effect. (2) Packing the ternary values in 2-bit memory. This method is easier to implement, but in that case the computational complexity of the ternary kernel is about 4 times that of binary kernels [ref1]. Thus, there is an accuracy-computation trade-off between binary or ternary models.
>
> [ref1] Fast matrix multiplication for binary and ternary CNNs on ARM CPU (ICPR 2022)
>
> **Q2**: How does quantization affect the quality of vision transformers?
>
> **A2**: Vision transformer (ViT) quantization could be a very good future direction to explore, which we have not studied yet. One difference between ViT and NLP transformers is that the feature map size of ViT is usually larger. This may pose new challenges in quantizing the ViT.
> In the literature, a DeiT-T [ref2] (a common ViT) with weights and activation quantized to 8-bit encounters 0.6% accuracy drop compared to full-precision DeiT-T [ref3]; quantizing DeiT-T to ~3-bit witnesses 3.2% drop [ref4]; and quantizing the weights to ternary and activation to 8-bit sees 5.6% accuracy drop on DeiT-T [ref5].
> Thus, we believe binarizing the vision transformer is a challenging task and may require combining the techniques found in this paper and additional adjustments customized for the ViTs.
>
> [ref2] Training data-efficient image transformers & distillation through attention (ICML 2021)
>
> [ref3] FQ-ViT: Post-Training Quantization for Fully Quantized Vision Transformer (IJCAI 2022)
>
> [ref4] Q-ViT: Fully Differentiable Quantization for Vision Transformer (Arxiv)
>
> [ref5] TerViT: An Efficient Ternary Vision Transformer (Arxiv)

---

### Author Response · Authors · 2022-08-02
**Summary of our rebuttal**

We thank the reviewers for their time and efforts in reviewing our paper. We are encouraged that the reviewers recognized that the proposed method is simple yet effective [Reviewer **2rah**], reduces the gap in GLUE score to the full-precision model by ~3X percentage points [Reviewer **4xBa**], improves the quality of Binarized BERT model significantly [Reviewer **2rah**] and delivers best practices for model replicability [Reviewer **Yoaz**].

Also, we appreciate reviewers’ constructive comments, e.g., more discussion on the previous work and hardware complexity [Reviewer **2rah**]; and insightful questions e.g., whether ternary quantization yields significant improvements compared with binary quantization [Reviewer **4xBa**], whether using other pre-training objectives will change the results [Reviewer **Yoaz**], etc.

In addition to the pointwise response below, we summarize our updates to the paper as follows:

[**Finer-grained comparison to the previous work**] We further provide more ablation studies to demonstrate the improvements brought by each component and discuss the crucial difference between the proposed method and the previous works, e.g., ReActNet and BiBERT.

[**Hardware complexity analysis**] We add the discussion of how to implement the {0, 1} activation binarization using the previous hardware implementation for {-1, 1} binarization without incurring any extra complexity.

We hope our responses below address all the concerns, and we thank all reviewers' efforts again.

---

### Meta-Review · Area_Chair_f9ca · 2022-08-27

**Recommendation:** Accept
**Confidence:** Certain

**Metareview:**

This paper proposes an innovative pipeline for quantizing transformers for extremely low precision (1-2) bits, while reducing the gap of previous methods to full precision by ~3X.

This result has important implications for resource-restricted inference, especially if memory is of concern, but 1-bit quantization has significant effect on inference latency as well.

This work reaches these strong results by careful normalization, separate quantization for non-negative activations and  a combinatorial optimization over various distillation paths.

Overall, the paper demonstrates an important albeit incremental advance in the field and is of general interest to the wider community, therefore I propose its acceptance at NeurIPS 2022.



**Award:**

No

---

### Decision · Program_Chairs · 2022-09-14

Accept